# Proteomic characterization and comparison of ram (*Ovis aries*) and buck (*Capra hircus*) spermatozoa proteome using a data independent acquisition mass spectometry (DIA-MS) approach

**Wen Zhu**[1]☯, **Xiao Cheng**[1]☯, **Chunhuan Ren**[1], **Jiahong Chen**[2], **Yan Zhang**[1], **Yale Chen**[1], **Xiaojiao Jia**[1], **Shijia Wang**[1], **Zhipeng Sun**[1], **Renzheng Zhang**[1], **Zijun Zhang**[1]*

**1** College of Animal Science and Technology, Anhui Agricultural University, Hefei, P. R. China, **2** New Rural Develop Research Institute, Anhui Agricultural University, Hefei, P. R. China

☯ These authors contributed equally to this work.
* zhangzijun@ahau.edu.cn

**Data Availability Statement:** All mass spectrometry proteomics data are available from the PRIDE database (accession number

## Abstract

Fresh semen is most commonly used in an artificial insemination of small ruminants, because of low fertility rates of frozen sperm. Generally, when developing and applying assisted reproductive technologies, sheep and goats are classified as one species. In order to optimize sperm cryopreservation protocols in sheep and goat, differences in sperm proteomes between ram and buck are necessary to investigate, which may contribute to differences in function and fertility of spermatozoa. In the current work, a data-independent acquisition-mass spectrometry proteomic approach was used to characterize and make a comparison of ram (*Ovis aries*) and buck (*Capra hircus*) sperm proteomes. A total of 2,109 proteins were identified in ram and buck spermatozoa, with 238 differentially abundant proteins. Proteins identified in ram and buck spermatozoa are mainly involved in metabolic pathways for generation of energy and diminishing oxidative stress. Specifically, there are greater abundance of spermatozoa proteins related to the immune protective and capacity activities in ram, while protein that inhibit sperm capacitation shows greater abundance in buck. Our results not only provide novel insights into the characteristics and potential activities of spermatozoa proteins, but also expand the potential direction for sperm cryopreservation in ram and buck.

## Introduction

Cryopreservation of spermatozoa is an important tool for breed improvement in small ruminants. However, frozen sperm is not commonly used in artificial insemination (AI) of small ruminants due to the low fertility rates [1]. Sheep (*Ovis. aries*) and goat (*Capra. hircus*) is a major food-animal group as well as a major source of wool and cashmere [2]. Generally, when

PXD014095). With reviewer account details: Username: reviewer92861@ebi.ac.uk Password: hB4YOBzO

**Funding:** This work was supported by National Key R&D program of China (No. 2018YFD0502001) awarded by Zijun Zhang, Key Program for Youth Science Foundation of Anhui Agricultural University (2018zd18) awarded by Wen Zhu, Introduced and Stable Program for the Talents of Anhui Agricultural University (yj2018-53) awarded by Wen Zhu, and from China Agriculture Research System (CARS-38) awarded by Zijun Zhang. The funders had no role in study design, data collection and analysis, decision to publish, or preparation of the manuscript.

**Competing interests:** The authors have declared that no competing interests exist.

developing and applying assisted reproductive technologies, sheep and goats are classified as one species. For example, most of the sperm cryopreservation procedures used for goat are extrapolated from those developed for the sheep species [3–5]. However, differences exist between sheep rams and goat bucks in regards to sperm morphometric [6] and concentration [7], which indicated various ability to undergo capacitation and fertility [8]. Spermatozoa response to the cryopreservation procedure was also different between rams and bucks, such as semen collection method [1]. In order to improve AI efficiency in sheep and goat, a deeper understanding of sperm biology is needed urgently.

Spermatozoa are highly specialized cells, which are inactive in transcription, proteomics-based technique is one of the great methods to understand the sperm molecular functions [9]. Over the last few years, comparative proteomics showed the ability to detect the difference and screen biomarkers as well as interesting candidates for further study among individuals or species, which may contribute to differences in fertility [10–11] and sperm function [12–14]. Furthermore, spermatozoa proteome characterization has been published for both sheep [15] and goat [7], however, there are no directly comparable between these two species. A comprehensive proteomic profiling between ram and buck spermatozoa is important for the identification of potential interesting proteins, with the end goal is to improve success rate of AI in these agriculturally important animals. Because of simultaneous high-throughput scanning and identifying all peptides within a given mass range, data-independent acquisition- mass spectrometry (DIA-MS) provides the possibility of overcoming limitations related to data dependent acquisition (DDA) [16]. As a highly promising MS-acquisition method, DIA has been successful used to investigate proteomic changes in plasma [17], cell [18], and tissue [19] in recent years.

To better investigate the sperm protein characteristics and estimation their differences, comparative analyses of sperm proteins between ram (*Ovis. aries*) and buck (*Capra. hircus*) are necessary. Thus, the objective of the present study was to systematically characterize and make a comparison of ram and buck spermatozoa proteome using a DIA-MS approach.

## Material and methods

### Experimental design and workflow

The experimental design and workflow are shown in Fig 1. The experimental design and workflow are shown in Fig 1. Semen samples were collected from Hu-sheep rams (n = 9) and Anhui white goat bucks (n = 9). Individual spermatozoa of each species were equally pooled into three biological samples.

### Chemicals

In this study, unless otherwise stated, the reagents were purchased from Sigma (St. Louis, MO, USA), and the diluents were prepared using Milli-Q water (Millipore Ibérica S.A., Barcelona, Spain).

### Animals

The Animal Care Committee of Anhui Agriculture University proved the use of animals for the experiment (Hefei, China). Mature Hu-sheep and Anhui white goat were housed at Anxin Husband Inc. (Fuyang, P.R. China) and fed 2 times daily at 06:00 and 18:00 h with free access to drinking water. Animals were fed a Chinese ryegrass-based diet with concentrate supplementation.

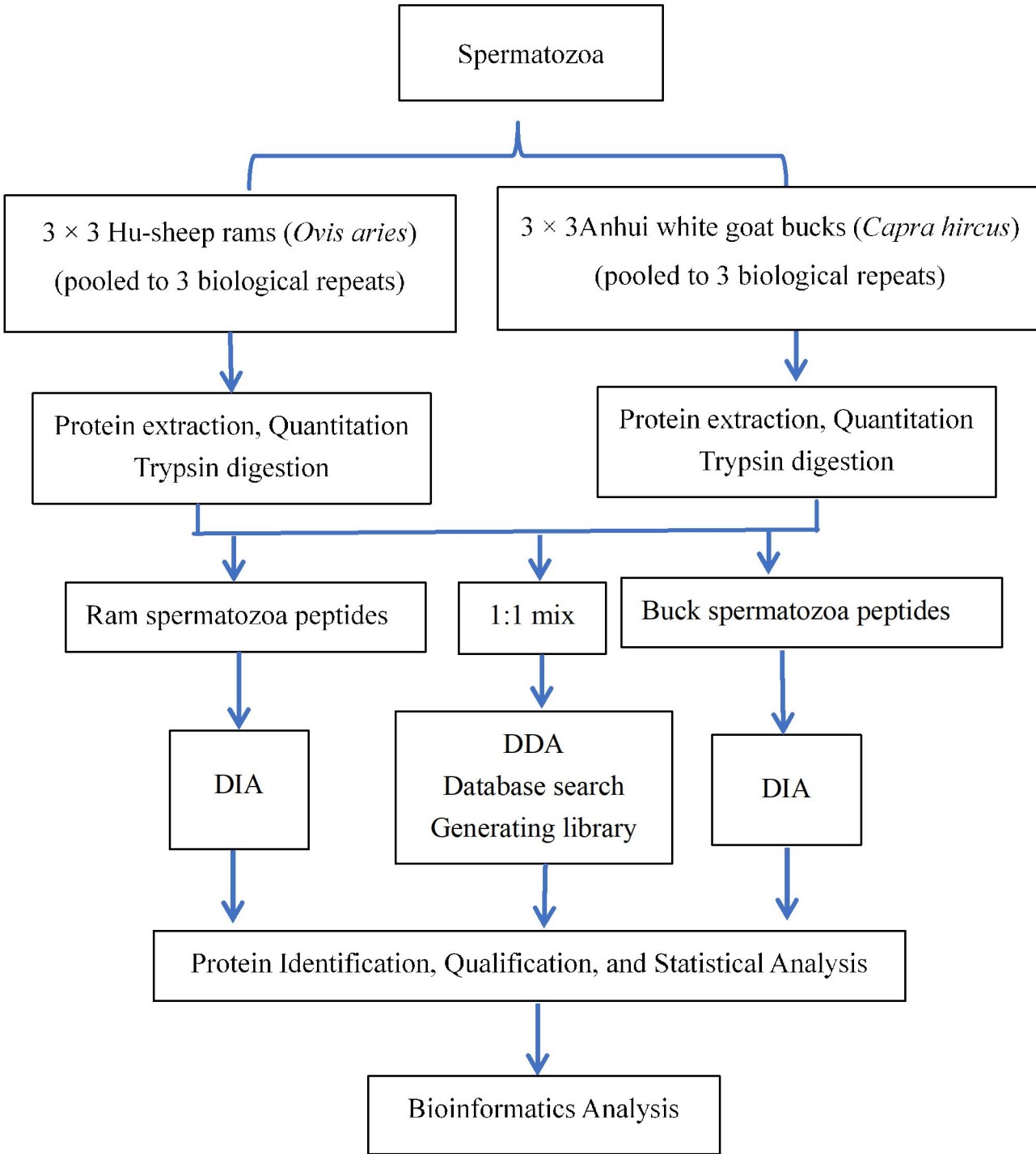

**Fig 1. Experimental design and work flow for the comparison of ram and buck spermatozoa using a data-independent acquisition-based mass spectrometry technology.**

## Collection and preparation of spermatozoa

Semen from 18 (9 rams, Hu-sheep, *Ovis aries*; and 9 bucks, Anhui white goat, *Capra hircus*) mature males (1.5 ± 0.25 years) were collected by artificial vagina (April 2017) using a

previously described method [20]. Ejaculates were assessed for volume, concentration and motility immediately by a CASA system (sperm class analyzer (SCA)-5.4.0.0; Microptic Supply, Barcelona, Spain) according to Mortimer et al. [21]. Ejaculates (n = 2/male) from each animal were pooled together, and then semen samples (1.70 ± 0.084 mL volume; 3.99 ± 0.52×108/ mL concentration; 89.5 ± 8.56% A+B grade motility) were divided into 3 groups randomly (n = 3) for each species. Spermatozoa was separated from seminal plasma by centrifugation (2,500 × g, 30 min, 4˚C), and supernatant was discarded, then the remainder of each sample was washed in PBS twice by centrifugation (2,500 × g, 10 min, 4˚C). The sperm pellets were stored -80˚C until further use.

## Protein extraction and digestion

The frozen sperm pellets were dissolved in lysis buffer (1mM Phenyl methane sulfonyl fluoride (Bio Basic Inc., Amherst, NY, USA), 2 mM ethylenediamine tetra acetic acid (Amersco, Burlington, MA, USA), 1% Protease Inhibitor Cocktail (Basel, Swiss, Roche)), oscillated on the Vortex Oscillator (Shanghai Damu Industrial Co., LTD, Shanghai, P. R. China), incubated on ice for 5 min. Then 10 mM dithiothreitol (DTT; Amersco, MA, USA) was added into the mixture and disrupted by tissue lysing machine (120 s, 50 HZ/s, Shanghai Jingxin Industrial Development Co., LTD, Shanghai, P. R. China). After centrifugation at 25,000 × g for 15 min at 4˚C, 10 mM DTT was added into the supernatant and kept at 56˚C for 1 h. Then, 55 mM iodoacetamide was added to the solution and incubated for 45 min at room temperature in the dark, and followed by mixed with four times volumes of cold acetone (Guangdong Shantou Xilong Chemical Co., LTD. Shantou, P.R. China), and stored at -20˚C for 2 h (This step was repeated three times). The supernatant was discarded after centrifugation (25,000 × g, 20 min, 4˚C), and the remaining debris was lysed in the above lysis buffer, sonicated (120 s, 50 HZ/s) by tissue lysing machine followed by centrifugation (25,000 × g, 4˚C, 20 min). Finally, the protein concentration of the supernatant was measured by the Bradford method [22]. The sperm protein was kept frozen at -80˚C until used.

For protein digestion, each protein sample (100 μg) were digested overnight with trypsin at 37˚C (trypsin: protein = 1:40 (v/v); Promega; Madison, WI, USA). Enzymatic peptides were then desalted by StrataX column and vacuum dried prior to MS.

## High pH reverse-phase separation

All samples were mixed equally (20 μg/sample), and 100 μg subsample was re-dissolved in 2 mL buffer A (5% acetonitrile (ACN); pH 9.8) and fractionated by an LC-20AB system (Shimadzu, Japan) connected to a reverse-phase Gemini C18 column (4.6 mm × 250 mm, 5 μm, Phenomenex, CA, American). The samples were subjected to the column and then eluted at a rate of 1mL/min: 5% (v/v) buffer B (95% ACN; pH 9.8) for 10 min, 5–35% (v/v) buffer B for 40 min, 35–95% (v/v) buffer B for 1 min, flow buffer B lasted 3 min and 5% buffer B equilibrated for10 min. The elution peak was monitored at a wavelength of 214 nm, the sample fractions were collected every 1 min. Then, components were combined into 10 fractions, and freeze-dried.

## Data dependent acquisition (DDA) and DIA analysis by nano-LC-MS/MS

All experiments were carried out with a Q Exactive HF mass spectrometer (Thermo Fisher Scientific) coupled with an Ultimate 3000 RSLCnano system (Thermo Fisher Scientific). A nano-LC column (150 μm × 25 cm, 1.8 μm, 100 Å) was packed in-house for peptide separation at a flow rate of 500 nl/min. For DDA analysis, the peptides were re-dissolved with buffer C (2% ACN; 0.1% formic acid (v/v)), and centrifuged (20,000 × g; 10 min; 4˚C). Then the supernatant

was loaded onto a trap column (300 μm × 5 mm, 5 μm, Thermo Scientific) and eluted with a set gradient, from in 5% (v/v) buffer D (98% CAN; 0.1%formic acid (v/v)) for 5 min, 5–35% (v/v) buffer D for 155 min, 35–80% (v/v) buffer D for 10 min, 80% (v/v) buffer D for 5 min, and 5% (v/v) buffer D for 4 min. The MS parameters were set as below: (1) MS: 350–1,500 scan range (m/z); 60,000 resolution; 3e6 AGC target; 50 ms maximum injection time (MIT); 30 loop count; 28 NCE; (2) HCD-MS/MS: 15,000 resolution; 1e5 AGC target; 100 ms MIT; charge exclusion, exclude 1, 7, 8, >8; filter dynamic exclusion duration 30 s; isolation window 2.0 m/z. For DIA analysis, the same nano-LC system and gradient was used as DDA analysis. The DIA MS parameters were set as below: (1) MS: 350–1,500 scan range (m/z); 20 ppm MS tolerance; 120,000 resolution; 3e6 AGC target; 50 ms MIT; 50 loop count; (2) HCD-MS/MS: 1.7 m/z isolation window; 30,000 resolution; 1e6 AGC target; automatic MIT; 50 loop count; filter dynamic exclusion duration 30s; stepped NCE: 22.5, 25, 27.5.

## Mass spectrometric raw data analysis

Raw data of DDA were processed and analyzed by MaxQuant (version 1.5.3.30). The identifications were filtered for no more than 1% FDR both on peptide and protein level. The DDA files were searched against an in-house Uniprot database of sheep and goat with 27,533 and 2,859 entries, respectively (03–2018). Search parameters and settings were as follows: (i) trypsin enzyme; (ii) 7 minimal peptide length; (iii) Carbamidomethyl as fixed modifications (C); (iv) oxidation (M) and acetyl (protein N-term) as the variable modifications. DIA were analyzed by Spectronaut Pulsar 11.0 (Biognosys AG), which uses the iRT peptides for retention time calibration [23]. The FDR was estimated with the mProphet scoring algorithm, and set to no more than 1% at peptide precursor level. Then, based on the target-decoy model applicable to SWATH-MS, to obtain quantitative results. All raw MS data have been deposited to the ProteomeXchange Consortium through the PRIDE partner repository (Identifier: PXD014095).

## Western blotting validation

Western blot was performed according to our previously described method [20]. Briefly, protein samples (60 ug protein per sample) from ram and buck spermatozoa were separated prior to detection with antibodies: CRP (1:1000), DEFB1(1:1000), COII (1:1000), and ALDH2 ((1:1000) from Abcam (Abcam, Cambridge, MA, USA); NUDT18 (1:800) from BIOSS (BIOSS, Beijing, P. R. China). Goat anti-rabbit (Abcam, Cambridge, MA, USA) (diluted at 1:5000) was used as the second antibody. Finally, blot was visualized and the band gray values were calculated.

## Bioinformatic analysis and statistical analyses

Identified proteins from spermatozoa were annotated and classified into pathway by gene ontology (GO) (http://david.abcc.ncifcrf.gov/home.jsp) and the kyoto encyclopedia of genes and genomes (KEGG) database (http://www.genome.jp/kegg/) database, respectively. Principal component analysis (PCA) of the quantified proteins were processed by Unscrambler software (Camo, version 9.8, Norway). Significant GO functions and pathways were examined within different expressed proteins (DEPs) with $p$- value $\leq$ 0.05. The protein-protein interaction (PPI) network of the difference proteins was analyzed by the web-tool STRING 11.0 (http://string-db.org).

Significance between ram and buck spermatozoa was determined by a Student's $t$-test in the MSstats, and $p < 0.05$ was considered significant. In order to increase the quantitative comparison validity, unique peptides $\geq$ 2 and the fold change $\geq$ 2 were selected. Western blot data were analyzed by a Student's $t$-test in SPSS (v.22.0, Chicago, IL, USA), where $p < 0.05$ was considered significant.

## Results

### Protein identification and quantification

DIA-MS requires an assay library containing all spectra peptides to be quantified. A spectral library consisting of 16,143 peptides belonging to 3,217 proteins were built in our study. The details of the proteins generated by DDA are list in S1 Table. Totally, 13,195 peptides, corresponding to 2,288 proteins (2,197 proteins for ram, 91 proteins for buck) were identified, and 2,109 proteins for further qualified. Details of protein identified and quantified by DIA are shown in S2 and S3 Tables, respectively.

For GO cellular component annotation, the most representative proteins were classified into cell, cell part, and organelle; For GO biological process annotations, the most represented were cellular process, single-organism process, and metabolic process; For GO molecular function annotations, the most prevalent represented were the binding and catalytic activity (Fig 2). Pathway analysis of all identified proteins from ram and buck were shown in S4 and S5 Tables, respectively. We noted that most proteins were involved in metabolic pathways.

### Principal component analysis

PCA of qualified spermatozoa proteins showed that samples from sheep and goat were in a separate cluster (Fig 3). The first two PCs explained 68.02% of the total variance, which could distinguish the two species.

### DEPs in ram and buck

Totally 238 significantly differential abundant sperm proteins were identified ($p < 0.05$, fold change $> 2$ or $< 0.5$), which included 166 up-regulated proteins and 72 down-regulated proteins in comparison of ram with buck spermatozoa. Detailed information of the DEPs is shown in S6 Table. In addition, the top 10 up- and down-regulated proteins with the highest differential abundance in the comparison of ram with buck spermatozoa are listed in Table 1.

### Validation of DEPs by western blot

We selected five proteins CRP, DEFB1, COII, NUDT18, and ALDH2 to be validated by Western blot (Fig 4). Validation of the five proteins by Western blot (Fig 5A) were consistent with the results of DIA proteomic analyses (Fig 5B), indicating that the proteomics data were highly reliable.

### GO classification and pathway enrichment of the DEPs

GO annotation was used to identify the functions of the DEPs between ram and buck spermatozoa (S7 Table). Among the 238 DEPs between ram and buck spermatozoa, 171 proteins had annotated functions and were classified into 47 functional groups (Fig 6A). Among them, the biological process accounted for 23 GO terms (the most representative were cellular process, metabolic process, and biological regulation), cellular component accounted for 15 GO terms (the most representative were cell, cell part, organelle, organelle part, membrane, and extracellular region), and molecular function accounted for 9 GO terms (the most representative were catalytic activity and binding).

According to the analysis of the enrichment of KEGG pathway, 16 pathways were assigned to DEPs in ram and buck (Fig 6B), and all enriched terms are shown in S8 Table. Most proteins were associated with microbial metabolism in diverse environments, valine, leucine and isoleucine degradation, glycine, serine and threonine metabolism, and other glycan degradation.

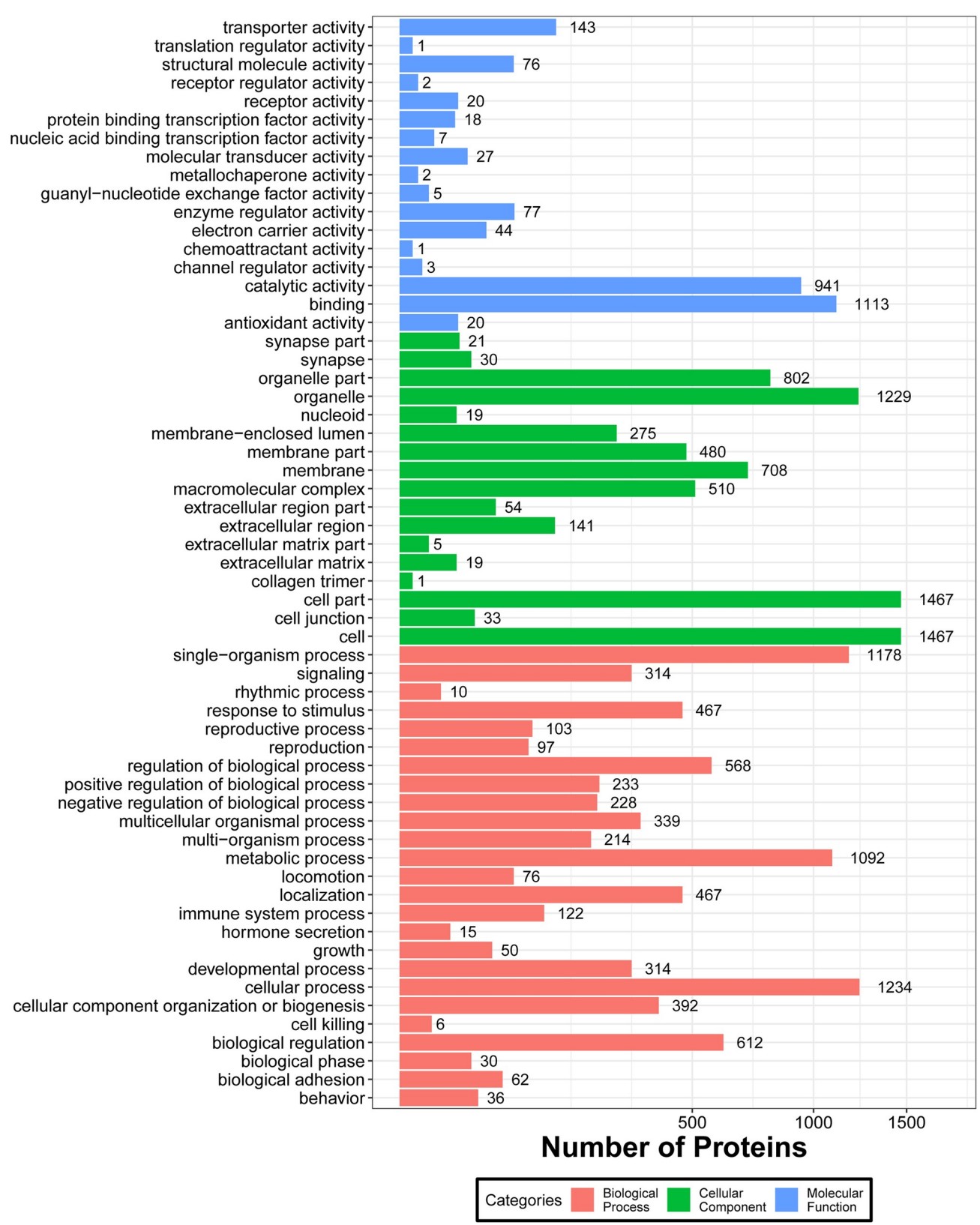

**Fig 2. Bar graph of gene ontology (GO) classification of all identified spermatozoa proteins by data-independent acquisition-based mass spectrometry.** The length shows the number of all differentially abundant proteins associated with the GO term.

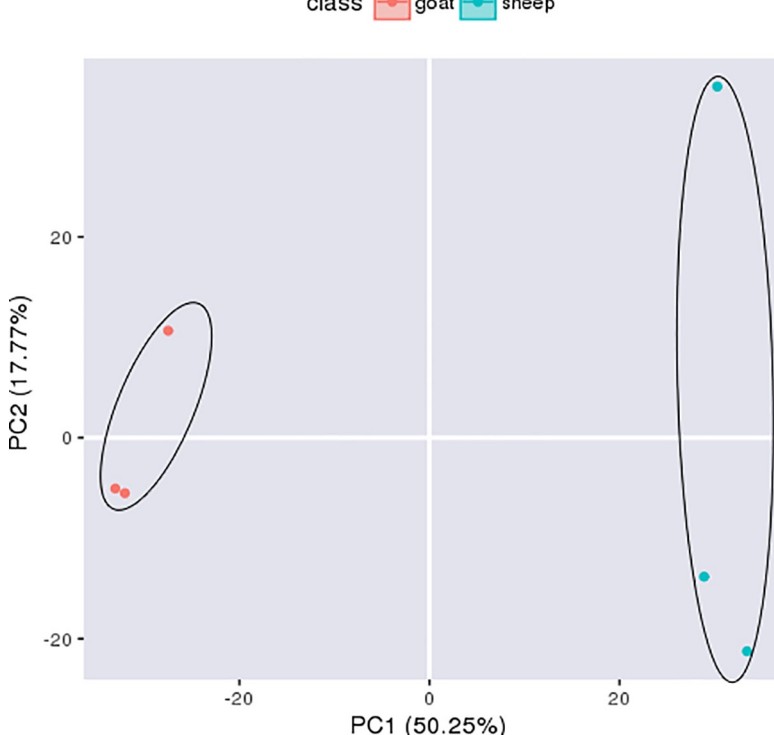

**Fig 3. Principal component analysis (PCA) scores plot of qualified spermatozoa proteins from ram and buck.**
Each symbol represents one sample, and group of samples defined by PCA are enclosed in circles.

**Table 1. The top 10 up- and down-regulated proteins with the highest differential abundance in ram comparing with buck spermatozoa.**

| Up-regulated | | | | | Down-regulated | | | | |
|---|---|---|---|---|---|---|---|---|---|
| Protein Accession | Protein Description | Gene Name | log$_2$FC[a] | P-value[b] | Protein Accession | Protein Description | Gene Name | log$_2$FC[a] | P-value[b] |
| W5PD71 | Pentaxin | CRP | 6.43 | 0.041 | W5PEC3 | Annexin | ANXA6 | -4.95 | 0.030 |
| W5PJJ9 | Uncharacterized | LOC101114850 | 5.96 | 0.009 | G1DGI1 | Serpine 2 protein | SERPINE2 | -4.65 | 0.004 |
| W5P1P8 | Beta-defensin 1 | DEFB1 | 5.94 | 0.009 | W5PLD7 | Nudix hydrolase 18 | NUDT18 | -3.34 | 0.012 |
| W5PHD0 | Uncharacterized protein | LOC101111693 | 5.79 | 0.005 | W5P6F3 | Phospholipid scramblase | NA | -2.81 | 0.044 |
| A0A0P0KLF3 | Cytochrome c oxidase subunit 2 | COII | 5.36 | 0.001 | I3WAE6 | HSP27 protein (Fragment) | HSP27 | -2.64 | 0.043 |
| W5P9A0 | Aldehyde dehydrogenase 5 family member A1 | ALDH5A1 | 4.40 | 0.004 | W5P220 | Glyoxylate reductase 1 homolog | GLYR1 | -2.57 | 0.037 |
| W5PEB3 | Uncharacterized | LOC101111505 | 3.73 | 0.005 | W5QBX4 | Solute carrier family 25 member 24 | SLC25A24 | -2.41 | 0.001 |
| W5PSZ8 | Triokinase and FMN cyclase | TKFC | 3.53 | 0.025 | W5QIV1 | Protein S100 | S100A11 | -2.39 | 0.012 |
| W5NU21 | Alpha-L-fucosidase, | FUCA2 | 3.18 | 0.008 | W5PHD3 | Sigma non-opioid intracellular receptor 1 | SIGMAR1 | -2.27 | 0.018 |
| W5PV74 | Acylaminoacyl-peptide hydrolase | APEH | 3.14 | 0.012 | W5PB83 | Serine hydroxymethyltransferase | SHMT2 | -2.21 | 0.050 |

[a]FC, Fold change

[b]p values were calculated using Student's t-Test.

## Proteins networks analysis

STRING analysis of the DEPs from ram and buck formulated color-coded networks which were largely based on their associations (Fig 7). The functional modules were mainly involved in microbial metabolism in diverse environments (ME3, ACLY, FBP1, PGK2, ACAA2, GOT2, and LDHAL6B) and other glycan degradation (ALDH2, ACAA2, HIBCH, and ACAD8). S9 Table showed the central functional modules based on the PPI networks.

## Discussion

### The characterization of identified/differential spermatozoa proteins

In the present study, a total of 2,109 sperm proteins from ram and buck were identified and quantified by DIA-MS proteomics. Ram sperm proteins have been identified in several studies (i.e., Merino ram, Small-tail Han ram, Dorset ram). For example, Pini et al. reported that a total of 685 proteins were identified in ejaculated ram spermatozoa with the most abundant proteins involved in metabolic pathways [15]. In another study, SWATH-MS was applied to Merino ram proteins of fresh and frozen spermatozoa, identified 1,154 proteins and uncovered 51 DEPs, respectively [24]. Additionally, 25 proteins with differentially abundant between fresh and freeze-thawed in Dorset ram spermatozoa were identified by two-dimensional electrophoresis [12]. Only a small number of buck sperm proteins were identified by two-dimensional electrophoresis [7]. The number of sperm proteins in ram and buck identified this study were larger than that of previous studies, which will expand our knowledge of the spermatozoa proteome.

Due to high energy demands for key cellular processes in sperm, such as motility, the acrosome reaction, and capacitation, energy production is the primary goal for spermatozoa protein [25]. Our finding showed that most abundant proteins identified in ram and buck spermatozoa were involved in metabolic pathways, which is consistent with Pini et al. [15]. Among these proteins some are directly involved in supply of ATP (e.g., ATP6, ATP5A1, ACLY, ND4, ND2, DN5, etc.).

### High abundance spermatozoa proteins in rams

We found pentaxin (CRP), β-defensin 1 (DEFB1), cytochrome c oxidase subunit 2 (COII), and Aldehyde dehydrogenase 5 family member A1(ALDH5A1) had higher abundances in ram

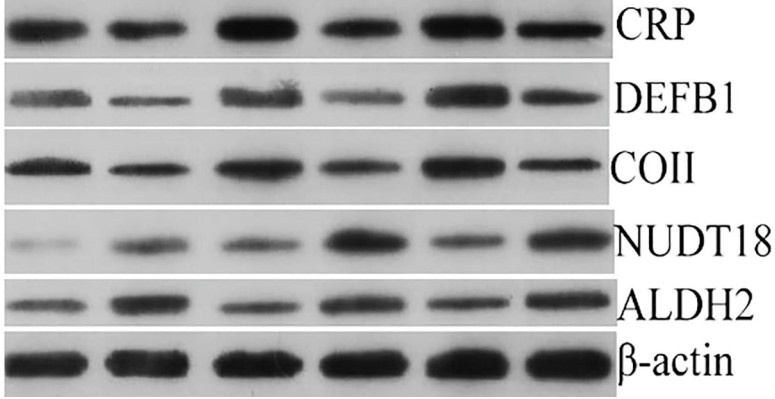

**Fig 4. Validation of selected different expressed proteins in ram and buck spermatozoa by Western blot.** The abundance of pentaxin (CRP), β-defensin 1(DEFB1), cytochrome c oxidase subunit 2 (COII), nucleoside diphosphate Type 18 (NUDT18), and aldehyde dehydrogenase 2 (ALDH2) proteins in ram and buck spermatozoa were analyzed by Western blot, with β-actin as an internal reference.

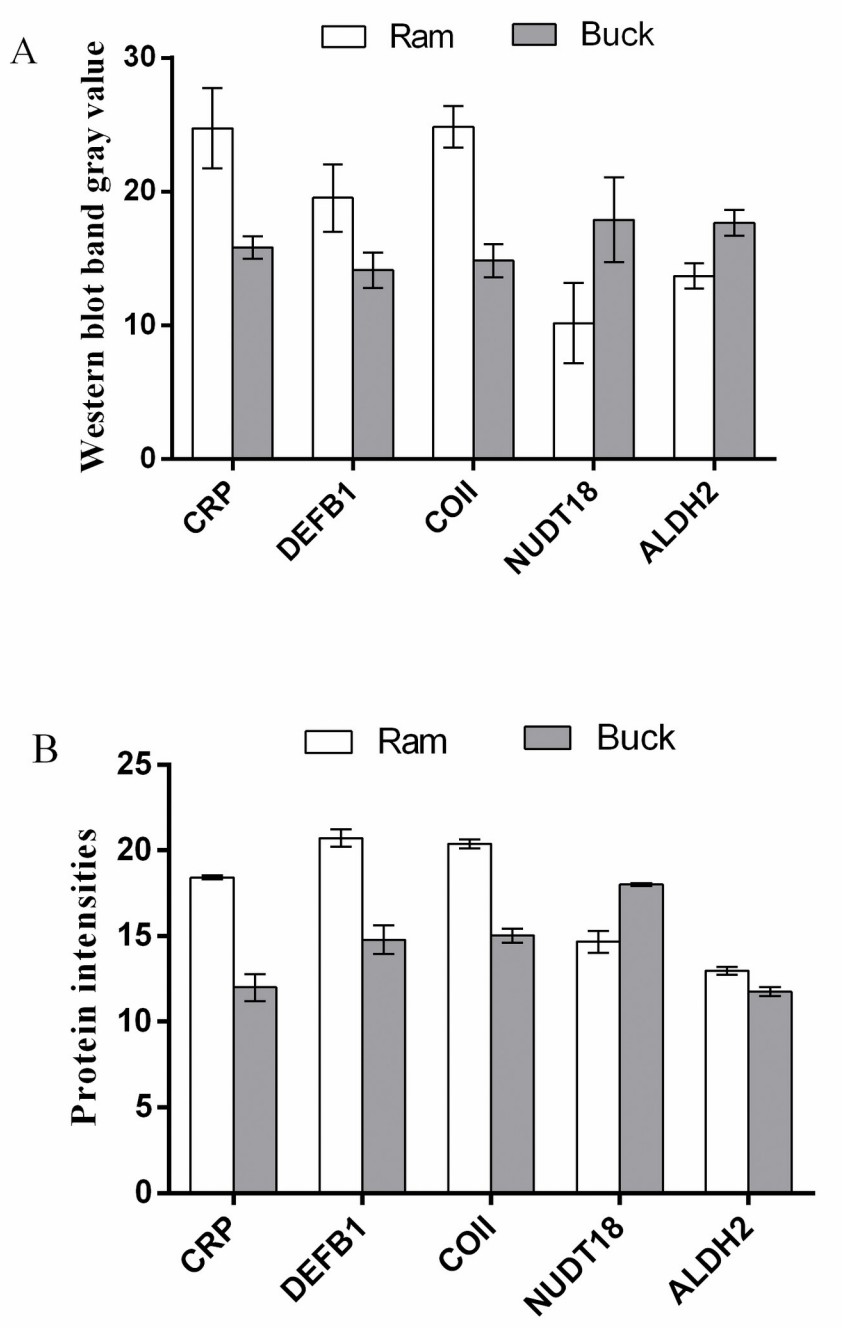

**Fig 5.** Expression levels of selected different expressed proteins as quantified by Western blotting (A) and DIA (B). The values are shown as mean ± SEM (n = 3 per group). All differences tested were statistically different ($P < 0.05$).

spermatozoa than buck spermatozoa in the current study. CRP has been reported as a conserved and species-specific sperm protein in ram [15]. CRP is produced by hepatocytes, which has pro- and anti-inflammatory activity, and multiple effects on the immune system [26]. β-defensins has been well known to act as signaling molecules in the immune system [27]. Studies have confirmed the expression of β-defensin genes (including DEFB1) in human, rat and ram epididymis [28–30], and Male mice with β-defensin gene knock-out were reported to be

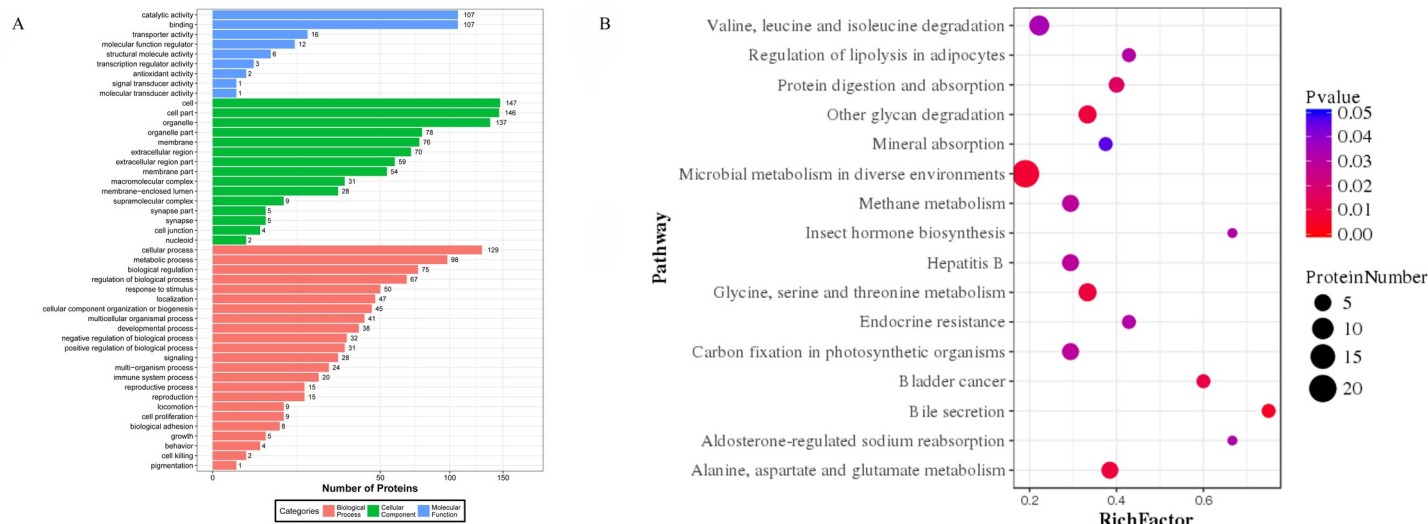

**Fig 6. Bar graph of gene ontology (GO) classification and bubble chart of Kyoto encyclopedia of genes and genomes (KEGG) pathway (B) analysis of differentially expressed proteins in ram vs. buck spermatozoa.** The bar length (A) and bubble size (B) show the number of all differentially abundant proteins associated with the GO and KEGG term.

infertile [29]. Higher expression of CRP and DEFB1 indicated an immune protection role for sperm within the female sheep reproductive tract. Mitochondria are important components of sperm, which provide energy supply for its motility, and sperm mitochondrial gene COII

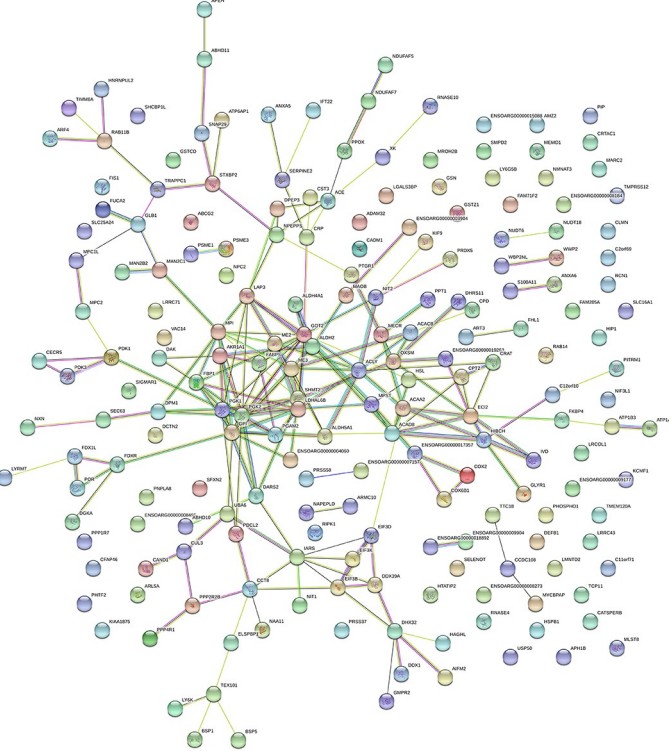

**Fig 7. STRING analysis of protein interaction network association of differentially expressed proteins in ram vs. buck spermatozoa.** Each node presents a protein; line colors present the types of evidence: purple lines from experimental determined, the blue lines from curated databases, and the yellow lines from text mining.

mutation is an important cause of low sperm motility in human [31]. As a member of aldehyde dehydrogenases family, ALDH5A1 is known to diminish oxidative stress [32–34]. Oxidative stress will inhibit mitochondrial respiration in freeze-thawed ram sperm [14]. The higher expression of ALDH5A1 abundance in ram spermatozoa may reduce the risk of oxidative stress during freeze-thawed process.

## Major spermatozoa proteins in bucks

Very few reports on buck sperm proteome were observed. In our study, the expression levels of annexin 6 (ANXA6), serpin peptidase inhibitor, clade E, member 2 (SERPINE2), and nucleoside diphosphate Type 18 (NUDT18) were the highest in the bucks among the spermatozoa proteins. As a calcium-dependent membrane binding protein, ANXAs is characterized by its ability to connect and bind to biological structures [35]. The deficiency of ANXA6 not only affects the calcium signaling but also affects the mitochondrial morphology in cells [36]. Expressions of annexins (ANXA1, ANXA2, and ANXA5) have been reported to be related with human sperm quality [37]. However, to date, the relationship between ANXA6 and sperm quality in bucks remains unclear and deserves further investigation. It is reported that SERPINE2 could inhibit sperm capacitation by blocking the cholesterol outflow of sperm plasma membranes and inhibiting the increase in the level of tyrosine phosphorylation in rat sperm protein [38]. Li et al. further confirmed that SERPINE2 could reversibly regulate mouse sperm from the state of capacitated to incapacitated [39]. Results indicated that capacitation inducers (such as BSA [36], cAMP analogs [40], and methyl-β-cyclodextrin [41]) or SERPINE2 inhibitor need to be added in buck sperm cryoprotectants for AI improvement. Reactive oxygen species (ROS) are byproducts of respiration using oxygen [42]. In the pool of DNA precursor, the guanine-containing nucleotides are converted into the oxidized forms of nucleotides by the action of ROS, such as 8-oxo-dGTP, which can be incorporated into DNA incorrectly [43]. NUDT18 could hydrolyze deoxyribonucleoside di- and tri-phosphates containing 8-oxoG into the monophosphate form in human cells, which could not be used for DNA synthesis [44]. Levels of ROS were significantly increased in freeze-thawed sperm, which indicated that the antioxidative system were disrupt during the freeze-thaw process [14]. Thus, higher expression of NUDT18 abundance in buck spermatozoa may involve in antioxidant stress during freeze-thawed process.

## Differences in spermatozoa proteins between ram and buck

As discussed above, there are variations in spermatozoa proteins between ram and buck, which contribute to the differential sperm characteristics. Differences in spermatozoa proteomes between ram and buck have been confirmed by the results of PCA in this study.

The abundance of 238 sperm proteins was significantly difference between ram and buck, and the other glycan degradation pathway was enriched. There were significant differences in 4 central functional modules (ALDH2, ACAA2, HIBCH, and ACAD8) associated with this pathway. In our study, it was found that the protein levels of acyltransferase 2 (ACAA2) and 3-hydroxyisobutyryl-CoA hydrolase (HIBCH) are higher in ram spermatozoa than that of buck, while the abundance of aldehyde dehydrogenase 2 (ALDH2) and acyl-coenzyme a dehydrogenase 8 (ACAD8) are lower. As discussed above, antioxidative system were disrupt during the sperm freeze-thaw process [14]. In the absence of the protective antioxidant factors, cells increase rates of lipid peroxidation, the formation of electrophilic aldehydes such as 4-hydroxynonenal (4HNE) are formed, the motility and membrane integrity are lost finally [45]. ALDH is an enzyme, which has responsibility for the removing 4-hydroxynonenal adducts from lipid membranes, and a strong correlation between ALDH2 expression and various

motility parameters of stallion spermatozoa were reported [46]. A higher abundant of ALDH2 in spermatozoa were also observed with high reproductive efficiency of Meishan boar [19]. To date, literature on the relative of ALDH2 expression with sperm functionality in buck is paucity, which is worthy of further exploration. ACAA2 and ACAD8 are proteins mainly catalyze dehydrogenation steps of β-oxidation processes in mitochondrial fatty acids catabolism [47–48], HIBCH involving in energy metabolism by regulating fat hydrolysis in obesity mice [49]. Taken above, these results indicated the important role of fatty acids catabolism in sperm.

Regulation of lipolysis in adipocytes pathway was another most representative pathway enriched based on DEPs between ram and buck spermatozoa. Lipolysis is the metabolic pathway, which has an important role in male fertility [50]. Hormone-sensitive lipase (HSL) proteins were enriched in the regulation of lipolysis in adipocytes pathway. In addition, we found that the levels of three HSL proteins were higher in ram spermatozoa that that of buck. HSL is an enzyme involved in fatty acid metabolism [51]. It has been reported that HSL was the main enzyme in the testis that hydrolyzes the cholesterol esters [52]. Cholesterol is the most abundant sterol of membrane properties in sperm cells that have the capacity to inhibit the capacitation [53]. It was reported that spermiogenesis ceased at the elongation phase of HSL-knockout mice [54]. Thus, higher abundance of HSL in ram spermatozoa probably represents higher capacity ability.

## Conclusions

In conclusion, the composition of spermatozoa proteins in ram and buck were investigated using DIA-MS proteomics. Proteins identified in ram and buck spermatozoa are mainly involved in metabolic pathways for generation of energy and diminishing oxidative stress. Specifically, the higher abundance of spermatozoa proteins in rams are associated with the immune protective and capacity activities, while protein that inhibit sperm capacitation shows greater abundance in buck. It is indicated that in order to achieve the high quality of frozen spermatozoa, cryopreservation of sperm in bucks should be different form rams. Furthermore, these difference abundant proteins might be research targets for improving AI.

## Supporting information

**S1 Table. Proteins identified via data dependent acquisition.**
(XLSX)

**S2 Table. Proteins identified via data independent acquisition.**
(XLSX)

**S3 Table. Proteins quantified via data independent acquisition.**
(XLSX)

**S4 Table. KEGG pathway analysis of the identified proteins in ram spermatozoa.**
(XLSX)

**S5 Table. KEGG pathway analysis of the identified proteins in buck spermatozoa.**
(XLSX)

**S6 Table. Differentially expressed proteins in ram vs. buck spermatozoa.**
(XLSX)

**S7 Table. GO classification of the differentially expressed proteins in ram vs. buck spermatozoa.**
(XLSX)

**S8 Table. KEGG pathway analysis of the differentially expressed proteins in ram vs. buck spermatozoa.**
(XLSX)

**S9 Table. The central functional modules based on the protein-protein interaction networks.**
(XLSX)

## Acknowledgments

We thank Miss. Mengnan Huang from BGI-Shenzhen Technology Co., Ltd for her technical support in mass spectroscopy.

## Author Contributions

**Conceptualization:** Wen Zhu, Zijun Zhang.

**Data curation:** Jiahong Chen.

**Formal analysis:** Wen Zhu, Chunhuan Ren.

**Funding acquisition:** Wen Zhu, Zijun Zhang.

**Investigation:** Wen Zhu, Xiao Cheng, Yan Zhang, Yale Chen, Zhipeng Sun, Renzheng Zhang.

**Project administration:** Wen Zhu, Chunhuan Ren.

**Resources:** Wen Zhu, Xiaojiao Jia, Shijia Wang.

**Validation:** Wen Zhu, Xiao Cheng.

**Visualization:** Wen Zhu.

**Writing – original draft:** Wen Zhu.

**Writing – review & editing:** Xiao Cheng, Chunhuan Ren, Jiahong Chen, Yan Zhang, Yale Chen, Xiaojiao Jia, Shijia Wang, Zhipeng Sun, Renzheng Zhang, Zijun Zhang.

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
