## [Decision Letter · Decision Letter 0]

20 Sep 2019

PONE-D-19-15817

Proteomic characterization and comparison of spermatozoa proteomes between ram (Ovis. aries) and buck (Capra. hircus) using a data-independent acquisition-mass spectrometry (DIA-MS)

PLOS ONE

Dear Dr Zhang,

Thank you for submitting your manuscript to PLOS ONE. After careful consideration, we feel that it has merit but does not fully meet PLOS ONE’s publication criteria as it currently stands. Therefore, we invite you to submit a revised version of the manuscript that addresses the points raised during the review process.

Most concerning to me personally when reading was the reviews was the comment about pseudoreplication. Note it will be important to submit high quality figures in all cases. Plos One provides a guide on how to do this. 

We would appreciate receiving your revised manuscript by Nov 04 2019 11:59PM. To enhance the reproducibility of your results, we recommend that if applicable you deposit your laboratory protocols in protocols.io, where a protocol can be assigned its own identifier (DOI) such that it can be cited independently in the future. For instructions see: http://journals.plos.org/plosone/s/submission-guidelines#loc-laboratory-protocols

We look forward to receiving your revised manuscript.

Kind regards,

Peter J. Hansen

Academic Editor

PLOS ONE

Journal Requirements:

https://www.sciencedirect.com/science/article/pii/S0378432016300537?via%3Dihub

https://pubs.acs.org/doi/10.1021/acs.jproteome.7b00369

https://www.sciencedirect.com/science/article/pii/S1874391915000068?via%3Dihub

https://pubs.acs.org/doi/10.1021/acs.jproteome.6b00530

https://www.sciencedirect.com/science/article/pii/S0027510717302014?via%3Dihub

In your revision ensure you cite all your sources (including your own works), and quote or rephrase any duplicated text outside the methods section. Further consideration is dependent on these concerns being addressed.

4. Please amend your list of authors on the manuscript to ensure that each author is correctly linked to an affiliation.

Authors’ affiliations should reflect the institution where the work was done (if authors moved subsequently, you can also list the new affiliation stating “current affiliation:….” as necessary).

Reviewers' comments:

Reviewer's Responses to Questions

**Comments to the Author**

1. Is the manuscript technically sound, and do the data support the conclusions?

Reviewer #1: Partly

Reviewer #2: Partly

2. Has the statistical analysis been performed appropriately and rigorously? 

Reviewer #1: I Don't Know

Reviewer #2: Yes

3. Have the authors made all data underlying the findings in their manuscript fully available?

Reviewer #1: Yes

Reviewer #2: Yes

4. Is the manuscript presented in an intelligible fashion and written in standard English?

Reviewer #1: Yes

Reviewer #2: No

5. Review Comments to the Author

Reviewer #1: The main aim of the present manuscript is to characterize sperm proteome in sheep (Hu-sheep breed. Ovis aries) and goat (Anhui white goat. Capra hircus) and identify differences among these proteomes that could help to detect proteins that could be used as potential markers of sperm fertility and functionality and also of sperm cryoresistance. The paper is, in my opinion, very interesting since the topic addressed is of great interest currently for the improvement of reproductive technologies outputs in small ruminants. For performing this study, the authors used 9 rams and 9 bucks for obtaining the ejaculates (one per male) determining three biological replicates (each one composed by three different ejaculate samples) in each species. Spermatozoa from all the ejaculates were obtained by centrifugation and processed for protein analysis by using a data-independent acquisition-mass spectometry approach. A total of 13, 195 peptides corresponding to 2,288 proteins (buck: 2,197 and ram: 91) which 2,109 were identified. Regarding the proteins present in different abundance, a total of 238 were detected, 166 up-regulated and 72-down regulated when sperm proteome of ram and buck were compared. The authors conclude that most of the proteins identified in bucks and rams are related to metabolic pathways. In addition, the authors suggest that the differences found between rams and buck’s sperm proteome should be taken into account for optimizing the results of cryopresevation protocols in these species. Although the results are interesting and could result of high applicability there some aspects that should be clarified by the authors before the manuscript was ready for its publication. The current version of the manuscript is, in my opinion, not relevant enough to be published and therefore, I recommend the publication of this manuscript after major revision.

Please see the specific comments below:

-Title: It seems that the title is incomplete. I would suggest to modify as follows: “Proteomic characterization and comparision of ram (Ovis aries) and buck (Capra hircus) spermatozoa proteome using a data independent acquisition mass spectometry (DIA-MS) approach”. Please, revise it and modify if convenient.

- Abstract: Page 2; line 25: “…in sperm protein proteomes between…”. Please, revise this sentence and modify if convenient.

- Introduction: Page 3; lines 57-58: Please modify this sentence to “ Furthermore, the sperm proteome that have been published…”. In addition, here and all along the manuscript I would suggest to use the word “species” instead of breed since goats and sheeps are two different livestock species. In my opinion the term species would be more appropiate.

- Introduction: Page 4; lines 72-74: The objective should be rewritten, in its present form it is not very clear.

- Material and Methods: Page 4; lines 78-82: Although a figure is included, the expalnation of the experimental design is very poor. An explained description of this experimental design should be included in the text.

- Material and Methods: General comment regarding protein analysis: In my opinion this section is also very poorly explained. The provided information is not informative enough and in addition it is not well organized. I would suggest to deeply modify this part of the manuscript in order to improve the its quality. In addition, ans statistical analysis section should be included.

- Results: Information about differences or repeatability among the three biological replicates should be included.

Results about GO annotation and pathways enrichment should be put together.

- Discussion: The discussion section is in my opinion too superficial and, although in some point the authors make very good reasoning, in general is of low relevance. This section should be revised and rewritten to make it more scientifically relevant. As I have nooted before, the work is interesting and with all the obtained information the authors could perform a very interesting and useful discussion.

- Conclusion: the conclusion is too long and in its present form is, again in my opinion, not adequate. I would recommend to rewritten it in order to make it more concise. In addition, a better justification of the authors for the usefulness of these proteins as potential additives for improving buck sperm cryopreservation ability are necessary

-Figures: Please, provide figures in a higher quality format.

Reviewer #2: In this paper the Authors provide descriptive data concerning the comparison of sperm proteins of ram and buck semen. More than 2 000 proteins were identified and 238 were found to differ in abundance between buck and ram. This finding is interesting and extends our knowledge about sperm proteins in ruminants.

Major critique

1. In my opinion this study is incomplete regarding seminal proteins. Only data for sperm but not seminal plasma proteome are provided. Seminal plasma proteins are found to interact with the surface of spermatozoa, so data both for seminal plasma and spermatozoa are vital for better understanding of sperm physiology.

2. There is a serious probability that pseudoreplicates instead of replicates were used in this study. Pseudoreplication occurs when observational data are pooled prior to statistical analysis and subsamples are incorrectly treated as true replicates for statistical analysis. The authors stated (L102) that semen samples were pooled.

3. Relationship of obtained results to cryopreservation is not provided. Cryopreservation is mentioned as the main justification for this study (L42-51). Cryopreservation experiments were not performed in this study and the importance of the data for cryopreservation is not discussed at all.

4. There is no validation of obtained results with other methods, for example Western blotting.

Specific comments

Introduction:

Line 20, 44 I would change “specie” to “species”

L21 Instead of “sperm protein proteomics” I would use “sperm proteins” or “sperm proteomes” throughout the MS.

L52 other “omics” techniques, especially transcriptomics, are used for studies of spermatozoa as well.

L98 age of animals? Basic description of animals should be provided.

L101 short description of CASA systems should be provided. Semen volume and sperm concentration should be provided.

L104 did you use inhibitor cocktail to prevent proteolysis?

L107 Which buffer did you use to wash the sperm?

L107 time for storage of sperm pellets at -80°C should be provided.

L118 and 131 please explain why you reduced and alkylate proteins two times before and after trypsin digestion

L125 remove “according to Bradford (repetition).

L134 Please explain what you mean by “the combined sample “used for high reverse-phase separation.

L139 10 min

- How did the authors ensure confidence in protein ID with only one unique peptide (Supplementary Tab. S1)?

- What was the exact fold change use for analysis? (Suplementary TableS5 showed Fold Change lower that 2).

L171 please provide the peptide mass tolerance and MS/MS tolerance during database searching

It would be interesting to see data of Table S4 independently for ram and buck.

Please improve the quality of Fig. 2.

Some categories of Fig. 3 seem not to be relevant to sperm physiology, for example “insect hormone synthesis”, “carbon fixation in photosynthetic organisms”,” methane metabolism”, and so on.

Discussion

The Authors did not discuss the obtained results.

It would be meaningful to compare published proteomes with the proteome of ram and buck presented in this study

The authors performed the GO classification and pathway enrichment of DEPs, but they did not discuss these results (Fig. 2A, 2B, Fig.3, table S6, table S7) the obtained results.

6. PLOS authors have the option to publish the peer review history of their article (what does this mean?). If published, this will include your full peer review and any attached files.

Reviewer #1: No

Reviewer #2: No

---

## [Author Response · Author response to Decision Letter 0]

2 Dec 2019

List of Corrections Made with the Comments of the editors and Reviewers

Editor comments:

AU: Revised as you suggested.

- We noticed you have some minor occurrence of overlapping text with the following previous publication(s), which needs to be addressed:

https://www.sciencedirect.com/science/article/pii/S0378432016300537?via%3Dihub

AU: We are sorry for our duplication, duplicated text (L36-38) has been addressed. 

https://pubs.acs.org/doi/10.1021/acs.jproteome.7b00369

AU: We are sorry for our duplication, duplicated text (L60-67) has been addressed. 

https://www.sciencedirect.com/science/article/pii/S1874391915000068?via%3Dihub

AU: We are sorry for our duplication, duplicated text (L56-60), (L289-291) has been addressed. 

https://pubs.acs.org/doi/10.1021/acs.jproteome.6b00530

AU: We are sorry for our duplication, duplicated text (L52-57) has been addressed. 

https://www.sciencedirect.com/science/article/pii/S0027510717302014?via%3Dihub

AU: We are sorry for our duplication, duplicated text (L342-345) has been addressed. 

In your revision ensure you cite all your sources (including your own works), and quote or rephrase any duplicated text outside the methods section. Further consideration is dependent on these concerns being addressed.

AU: Revised as you suggested, all duplicated text have been addressed. 

- We note that you have included the phrase “data not shown” in your manuscript. Unfortunately, this does not meet our data sharing requirements. PLOS does not permit references to inaccessible data. We require that authors provide all relevant data within the paper, Supporting Information files, or in an acceptable, public repository. Please add a citation to support this phrase or upload the data that corresponds with these findings to a stable repository (such as Figshare or Dryad) and provide and URLs, DOIs, or accession numbers that may be used to access these data. Or, if the data are not a core part of the research being presented in your study, we ask that you remove the phrase that refers to these data.

AU: Revised as you suggested, more details about semen characters were added in the manuscript. 

And the raw mass spectrometry proteomics data have been deposited to the ProteomeXchange Consortium via the PRIDE partner repository with the dataset identifier PXD014095. 

- Please amend your list of authors on the manuscript to ensure that each author is correctly linked to an affiliation.

Authors’ affiliations should reflect the institution where the work was done (if authors moved subsequently, you can also list the new affiliation stating “current affiliation:….” as necessary).

AU: Yes, we are sure that each author is correctly linked to the affiliation.

Reviewer #1 comments:

General comment:

The main aim of the present manuscript is to characterize sperm proteome in sheep (Hu-sheep breed. Ovis aries) and goat (Anhui white goat. Capra hircus) and identify differences among these proteomes that could help to detect proteins that could be used as potential markers of sperm fertility and functionality and also of sperm cry resistance. The paper is, in my opinion, very interesting since the topic addressed is of great interest currently for the improvement of reproductive technologies outputs in small ruminants. For performing this study, the authors used 9 rams and 9 bucks for obtaining the ejaculates (one per male) determining three biological replicates (each one composed by three different ejaculate samples) in each species. Spermatozoa from all the ejaculates were obtained by centrifugation and processed for protein analysis by using a data-independent acquisition-mass spectrometry approach. A total of 13, 195 peptides corresponding to 2,288 proteins (buck: 2,197 and ram: 91) which 2,109 were identified. Regarding the proteins present in different abundance, a total of 238 were detected, 166 up-regulated and 72-down regulated when sperm proteome of ram and buck were compared. The authors conclude that most of the proteins identified in bucks and rams are related to metabolic pathways. In addition, the authors suggest that the differences found between rams and buck’s sperm proteome should be taken into account for optimizing the results of cryopreservation protocols in these species. Although the results are interesting and could result of high applicability there some aspects that should be clarified by the authors before the manuscript was ready for its publication. The current version of the manuscript is, in my opinion, not relevant enough to be published and therefore, I recommend the publication of this manuscript after major revision.

AU: Thank you very much for your positive and valuable comments on our work, we have revised our manuscript according to your valuable suggestions. 

Special comments:

-Title: It seems that the title is incomplete. I would suggest to modify as follows: “Proteomic characterization and comparison of ram (Ovis aries) and buck (Capra hircus) spermatozoa proteome using a data independent acquisition mass spectometry (DIA-MS) approach”. Please, revise it and modify if convenient.

AU: Thank you for your valuable suggestion, and revised as you suggested.

- Abstract: Page 2; line 25: “…in sperm protein proteomes between…”. Please, revise this sentence and modify if convenient.

AU: Revised as you suggested. The sentence has been rewritten. 

- Introduction: Page 3; lines 57-58: Please modify this sentence to “Furthermore, the sperm proteome that have been published…”. In addition, here and all along the manuscript I would suggest to use the word “species” instead of breed since goats and sheeps are two different livestock species. In my opinion the term species would be more appropiate.

AU: Thanks a lot for your suggestion, the sentence has been rewritten, and the word “species” was used instead of breed in the whole manuscript. 

- Introduction: Page 4; lines 72-74: The objective should be rewritten, in its present form it is not very clear.

AU: Revised as you suggested. The sentence has been rewritten.

- Material and Methods: Page 4; lines 78-82: Although a figure is included, the explanation of the experimental design is very poor. An explained description of this experimental design should be included in the text.

AU: Revised as you suggested, the explanation of the experimental design has been added. 

- Material and Methods: General comment regarding protein analysis: In my opinion this section is also very poorly explained. The provided information is not informative enough and in addition it is not well organized. I would suggest to deeply modify this part of the manuscript in order to improve the its quality. In addition, and statistical analysis section should be included.

AU: Revised as you suggested, the M&M section (especially for protein analysis) has been rewritten and reorganized, and more details has been added including statistical analysis section.

- Results: Information about differences or repeatability among the three biological replicates should be included.

AU: Revised as you suggested, PCA analysis were added to elucidate the differences or repeatability among the three biological replicates (Fig 3). 

-Results about GO annotation and pathways enrichment should be put together.

AU: Revised as you suggested, GO annotation and pathways enrichment for different expressed proteins were put together (Fig 6). 

- Discussion: The discussion section is in my opinion too superficial and, although in some point the authors make very good reasoning, in general is of low relevance. This section should be revised and rewritten to make it more scientifically relevant. As I have nooted before, the work is interesting and with all the obtained information the authors could perform a very interesting and useful discussion.

AU: Revised as you and reviewers suggested. Discussion has been extended (yellow-highlighted in part of Discussions).

- Conclusion: the conclusion is too long and in its present form is, again in my opinion, not adequate. I would recommend to rewritten it in order to make it more concise. In addition, a better justification of the authors for the usefulness of these proteins as potential additives for improving buck sperm cryopreservation ability are necessary.

AU: Revised as you and reviewers suggested. Conclusion has been rewritten. 

-Figures: Please, provide figures in a higher quality format.

AU: Revised as you suggested.

Reviewer #2 comments:

General comment:

In this paper the Authors provide descriptive data concerning the comparison of sperm proteins of ram and buck semen. More than 2 000 proteins were identified and 238 were found to differ in abundance between buck and ram. This finding is interesting and extends our knowledge about sperm proteins in ruminants.

AU: Thank you very much for your positive and valuable comments on our work, we have studied your comments carefully and have made correction and revisions which was yellow-highlighted in the revised version. 

Major critique

-In my opinion this study is incomplete regarding seminal proteins. Only data for sperm but not seminal plasma proteome are provided. Seminal plasma proteins are found to interact with the surface of spermatozoa, so data both for seminal plasma and spermatozoa are vital for better understanding of sperm physiology.

AU: Yes, as you said seminal plasma proteins are essential for sperm function. We have also characterized and made comparison of ram (Ovis aries) and buck (Capra hircus) seminal plasma proteome, and the manuscript is preparing. In this manuscript, we focus on spermatozoa proteome. 

-There is a serious probability that pseudo replicates instead of replicates were used in this study. Pseudo replication occurs when observational data are pooled prior to statistical analysis and subsamples are incorrectly treated as true replicates for statistical analysis. The authors stated (L102) that semen samples were pooled.

AU: Yes, as you said, pseudo replication occurs when observational data are pooled prior to statistical analysis. In this experiment, pseudo replication would not exit. In this study, we used 9 rams and 9 bucks for obtaining the semen samples (2 ejaculates/animal) determining three biological replicates (each one composed by three different animal samples, n = 3) in each species. Semen samples within each male were pooled to obtain a representative semen sample. 

-Relationship of obtained results to cryopreservation is not provided. Cryopreservation is mentioned as the main justification for this study (L42-51). Cryopreservation experiments were not performed in this study and the importance of the data for cryopreservation is not discussed at all.

AU: Comprehensive proteomic profiling between ram and buck spermatozoa was performed in the study, with the end goal of improving AI success in ram and buck. Cryopreservation was an important part of AI, and cryopreservation mentioned in the introduction was used to justify ram and buck should not be lumped together in AI technologies. The objective of this study was to systematically characterize and make a comparison of ram and buck spermatozoa proteome, thus, cryopreservation experiments were not performed in this study.

-There is no validation of obtained results with other methods, for example Western blotting.

AU: Revised as you suggested. Expression of some DEPs were validated by Western blot (Results are shown in Fig 4 and Fig 5A). 

Special comments:

-Introduction:

Line 20, 44 I would change “specie” to “species”

AU: Revised as you suggested.

-L21 Instead of “sperm protein proteomics” I would use “sperm proteins” or “sperm proteomes” throughout the MS.

AU: Revised as you suggested.

-L52 other “omics” techniques, especially transcriptomics, are used for studies of spermatozoa as well.

AU: We are sorry for such arbitrary conclusion; the sentence has been rewritten. 

-L98 age of animals? Basic description of animals should be provided.

AU: Revised as you suggested, ages of animals were provided. 

-L101 short description of CASA systems should be provided. Semen volume and sperm concentration should be provided.

AU: Revised as you suggested, description of CASA systems and semen volume and sperm concentration has been provided.

-L104 did you use inhibitor cocktail to prevent proteolysis?

AU: For semen samples, protease inhibitor cocktail was not used. In our study, protease inhibitor cocktail was used to prevent proteolysis during protein extraction. 

-L107 Which buffer did you use to wash the sperm?

AU: Sperm was washed in PBS and the detail was added in the manuscript. 

-L107 time for storage of sperm pellets at -80°C should be provided.

AU: Due to the working time, sperm pellets were stored -80 °C one night (about 10 hours). 

-L118 and 131 please explain why you reduced and alkylate proteins two times before and after trypsin digestion

AU: We are sorry for our mistake in writing, proteins were only reduced and alkylated one time, description about “reduced and alkylate proteins” in L131 has been deleted. 

-L125 remove “according to Bradford (repetition).

AU: Revised as you suggested. 

-L134 Please explain what you mean by “the combined sample “used for high reverse-phase separation.

AU: Sorry for our ambiguous description, all samples were mixed equally (20 μg/sample), and 100 μg subsample was used for DDA analysis. The sentence has been rewritten. 

-L139 10 min

AU: Revised as you suggested. 

- How did the authors ensure confidence in protein ID with only one unique peptide (Supplementary Tab. S1)?

AU: First, to generate the final spectral library, raw data of DDA were processed and analyzed by MaxQuant (version 1.5.3.30), and the identifications were filtered for no more than 1% FDR on peptide and protein level. Then, DIA data were analyzed by Spectronaut Pulsar 11.0 (Biognosys AG), based on the target-decoy model applicable to SWATH-MS to obtain quantitative results, and the FDR was estimated with the mProphet scoring algorithm, and set to no more than 1% at peptide precursor level.

- What was the exact fold change use for analysis? (Suplementary TableS5 showed Fold Change lower that 2).

AU: Fold changes ≥ 2.0 were used for analysis, values showed in Supplementary TableS5 were log2FC, not the exact fold change value. 

-L171 please provide the peptide mass tolerance and MS/MS tolerance during database searching

AU: MS/MS tolerance: 20 ppm, and detail was added. 

-It would be interesting to see data of Table S4 independently for ram and buck.

AU: Revised as you suggested, they were separated. KEGG pathway analysis of the identified proteins in sheep and goat spermatozoa are shown in S4Table and S5Table, respectively. 

-Please improve the quality of Fig. 2.

AU: Revised as you suggested. 

-Some categories of Fig. 3 seem not to be relevant to sperm physiology, for example “insect hormone synthesis”, “carbon fixation in photosynthetic organisms”,” methane metabolism”, and so on.

AU: Yes, you are right, some categories of Fig.6 were not relevant to sperm physiology, this phenomenon has also been found in other experiments. As we known, KEGG pathways were enriched based on the known function of differentially expressed proteins. To date, sperm protein functions in ram and buck were not welled studied, which leading to the above result. 

Discussion

-The Authors did not discuss the obtained results.

It would be meaningful to compare published proteomes with the proteome of ram and buck presented in this study

-The authors performed the GO classification and pathway enrichment of DEPs, but they did not discuss these results (Fig. 2A, 2B, Fig.3, table S6, table S7) the obtained results.

AU: Revised as you and reviewers suggested. Discussion has been extended (yellow-highlighted in part of Discussions).

Once again, thank you very much for your comments and suggestions.

---

## [Decision Letter · Decision Letter 1]

31 Dec 2019

PONE-D-19-15817R1

Proteomic characterization and comparison of ram (Ovis aries) and buck (Capra hircus) spermatozoa proteome using a data independent acquisition mass spectometry (DIA-MS) approach

PLOS ONE

Dear Dr Zhang,

Thank you for submitting your manuscript to PLOS ONE. After careful consideration, we feel that it has merit but does not fully meet PLOS ONE’s publication criteria as it currently stands. Therefore, we invite you to submit a revised version of the manuscript that addresses the points raised during the review process.

Please make sure to handle all remaining issues identified by the reviewer. They are important points. We would appreciate receiving your revised manuscript by Feb 14 2020 11:59PM. To enhance the reproducibility of your results, we recommend that if applicable you deposit your laboratory protocols in protocols.io, where a protocol can be assigned its own identifier (DOI) such that it can be cited independently in the future. For instructions see: http://journals.plos.org/plosone/s/submission-guidelines#loc-laboratory-protocols

We look forward to receiving your revised manuscript.

Kind regards,

Peter J. Hansen

Academic Editor

PLOS ONE

Reviewers' comments:

Reviewer's Responses to Questions

**Comments to the Author**

1. If the authors have adequately addressed your comments raised in a previous round of review and you feel that this manuscript is now acceptable for publication, you may indicate that here to bypass the “Comments to the Author” section, enter your conflict of interest statement in the “Confidential to Editor” section, and submit your "Accept" recommendation.

Reviewer #1: (No Response)

Reviewer #2: All comments have been addressed

2. Is the manuscript technically sound, and do the data support the conclusions?

Reviewer #1: Yes

Reviewer #2: Yes

3. Has the statistical analysis been performed appropriately and rigorously? 

Reviewer #1: Yes

Reviewer #2: Yes

4. Have the authors made all data underlying the findings in their manuscript fully available?

Reviewer #1: Yes

Reviewer #2: Yes

5. Is the manuscript presented in an intelligible fashion and written in standard English?

Reviewer #1: Yes

Reviewer #2: Yes

6. Review Comments to the Author

Reviewer #1: The authors have addressed almost all the modifications suggested. However, in the present version of the manuscript there are some aspects that still need to be improved.

The explanation of the experimental design is still very poor and although the figure 1 can help, in my opinion is not enough.

In the current version of the manuscript I have not found the figure legends for the figures provided.

Only legends for figure 1 and 2 are provided in the main text of the manuscript. This is nor the appropriate place and, in addition, the explanations are not adequate being poor and not informative enough.

The quality of some figures is still not good, as for example figure 1 and 2. Please, revise it again and try to improve if possible.

Finally, I recommend the publication of this manuscript after minor revision.

Reviewer #2: The Authors have complied with most of my remarks which led to the improvement of the overall quality and sound of the paper. Therefore, I can now recommend the MS for publication.

7. PLOS authors have the option to publish the peer review history of their article (what does this mean?). If published, this will include your full peer review and any attached files.

Reviewer #1: No

Reviewer #2: No

---

## [Author Response · Author response to Decision Letter 1]

15 Jan 2020

List of Corrections Made with the Comments of Reviewers

Reviewer #1 comments:

-The authors have addressed almost all the modifications suggested. However, in the present version of the manuscript there are some aspects that still need to be improved.

AU: Thanks for your valuable comments on our work, and we have studied comments carefully and have made correction and revisions which was yellow-highlighted in the revised version.

-The explanation of the experimental design is still very poor and although the figure 1 can help, in my opinion is not enough. 

AU: The explanation of the experimental design has been revised. The explanation of the experimental design was revised as follows: the experimental design and workflow are shown in Fig 1. Semen samples were collected from Hu-sheep rams (n = 9) and Anhui white goat bucks (n = 9). Individual spermatozoa of each species were equally pooled into three biological samples. 

-In the current version of the manuscript I have not found the figure legends for the figures provided.

-Only legends for figure 1 and 2 are provided in the main text of the manuscript. This is nor the appropriate place and, in addition, the explanations are not adequate being poor and not informative enough. 

AU: All figure legends has been revised and shown in the revised version of the manuscript. According to the guide-for-authors, each figure caption should appear directly after the paragraph in which they are first cited. So, all the figure legends were put in in the main text of the manuscript. 

-The quality of some figures is still not good, as for example figure 1 and 2. Please, revise it again and try to improve if possible.

AU: Revised as you suggested. Resolution of figure 1 and 2 have been improved, and all figures have been uploaded to the Preflight Analysis and Conversion Engine (PACE) digital diagnostic tool to ensure that figures meet PLOS requirements. However, it is strange that the uploaded figures are very clear, but the ones presented in the draft are not good. 

-Finally, I recommend the publication of this manuscript after minor revision.

AU: Revised as you suggested. 

Once again, thank you very much for your comments and suggestions. 

Reviewer #2 comments:

The Authors have complied with most of my remarks which led to the improvement of the overall quality and sound of the paper. Therefore, I can now recommend the MS for publication.

AU: Thank you very much for your positive and valuable comments on our work.

---

## [Editor Report · Decision Letter 2]

22 Jan 2020

Proteomic characterization and comparison of ram (Ovis aries) and buck (Capra hircus) spermatozoa proteome using a data independent acquisition mass spectometry (DIA-MS) approach

PONE-D-19-15817R2

Dear Dr. Zhang,

We are pleased to inform you that your manuscript has been judged scientifically suitable for publication and will be formally accepted for publication once it complies with all outstanding technical requirements.

With kind regards,

Peter J. Hansen

Academic Editor

PLOS ONE
---

## [Editor Report · Acceptance letter]

5 Feb 2020

PONE-D-19-15817R2 

Proteomic characterization and comparison of ram (*Ovis aries*) and buck (*Capra hircus*) spermatozoa proteome using a data independent acquisition mass spectometry (DIA-MS) approach 

Dear Dr. Zhang:

I am pleased to inform you that your manuscript has been deemed suitable for publication in PLOS ONE. Congratulations! Your manuscript is now with our production department. 

With kind regards,

on behalf of

Dr. Peter J. Hansen 

Academic Editor

PLOS ONE